# CBQ: Cross-Block Quantization for Large Language Models

**Xin Ding**[1][*][†]   **Xiaoyu Liu**[1][*][†]   **Zhijun Tu**[2]   **Yun Zhang**[3]   **Wei Li**[2]   **Jie Hu**[2]
**Hanting Chen**[2]   **Yehui Tang**[2]   **Zhiwei Xiong**[1]   **Baoqun Yin**[1][‡]   **Yunhe Wang**[2][‡]
[1]University of Science and Technology of China   [2] Huawei Noah's Ark Lab
[3] Hong Kong University of Science and Technology (GZ)

## Abstract

Post-training quantization (PTQ) has played a pivotal role in compressing large language models (LLMs) at ultra-low costs. Although current PTQ methods have achieved promising results by addressing outliers and employing layer- or block-wise loss optimization techniques, they still suffer from significant performance degradation at ultra-low bits precision. To dissect this issue, we conducted an in-depth analysis of quantization errors specific to LLMs and surprisingly discovered that, unlike traditional sources of quantization errors, the growing number of model parameters, combined with the reduction in quantization bits, intensifies inter-layer and intra-layer dependencies, which severely impact quantization accuracy. This finding highlights a critical challenge in quantizing LLMs. To address this, we propose CBQ, a cross-block reconstruction-based PTQ method for LLMs. CBQ leverages a cross-block dependency to establish long-range dependencies across multiple blocks and integrates an adaptive LoRA-Rounding technique to manage intra-layer dependencies. To further enhance performance, CBQ incorporates a coarse-to-fine pre-processing mechanism for processing weights and activations. Extensive experiments show that CBQ achieves superior low-bit quantization (W4A4, W4A8, W2A16) and outperforms existing state-of-the-art methods across various LLMs and datasets. Notably, CBQ only takes 4.3 hours to quantize a weight-only quantization of a 4-bit LLAMA1-65B model, achieving a commendable trade off between performance and efficiency.

## 1 Introduction

Large language models (LLMs) (Wei et al. (2022a); Radford et al.; Zhang et al.; Brown et al. (2020b); Dettmers et al. (2022)), have sparked immense academic and industrial interest owing to their remarkable performance in handling complex natural languages tasks (Hendrycks et al. (2020b); Bisk et al. (2020b); He et al. (2017); Ainslie et al. (2023); Liu et al. (2024b)). During to significant computational resources for inference and deployment, the post-training quantization (PTQ) technique (Choi et al. (2018); Frantar et al. (2022a); Nagel et al. (2019); Wei et al. (2023); Li et al. (2025)) operating with limited calibration data and computational resources is more in demand for compressing LLMs.

Existing PTQ methods typically optimize models on a layer or block basis, addressing outliers (Wei et al. (2022b; 2023); Chee et al. (2024); Liu et al. (2024a)) and employing first- or second-order optimization techniques (predominantly optimizing models on a layer-by-layer or block-by-block basis) (Shao et al. (2023); Frantar et al. (2022b); Liu et al. (2023a)). However, these approaches often suffer from significant performance degradation, particularly in low-bit settings such as W2A16 and W4A4, as illustrated in Table 1, due to inherent limitations. Previous work, like AdaRound (Nagel et al. (2020)), analyzed rounding errors and showed that simple rounding is not always the optimal quantization strategy, greatly improving quantization for CNNs. This inspired us to analyze quantization loss for LLMs, comparing high-bit and low-bit scenarios. We found that in low-bit quantization,

---

[*]Equal Contribution
[†]This work was done during Xin Ding and Xiaoyu Liu's internship at Huawei Noah's Ark Lab
[‡]Corresponding author:bqyin@ustc.edu.cn, yunhe.wang@huawei.com

intra-layer and inter-layer dependencies within models become more pronounced, especially as model size increases. This indicates that previous methods, whether focused on optimizing quantization parameters within a layer or block through first- or second-order techniques, or on refining rounding errors, fall short of achieving optimal outcomes. Instead, it is essential to fully account for the inter-layer and intra-layer relationships.

To address this, we propose CBQ, a cross-block reconstruction-based PTQ method tailored for LLMs, surpassing traditional layer-wise and block-wise reconstruction techniques. CBQ introduces a cross-block dependency (CBD) into block-wise reconstruction, maintaining the integrity of the model's internal dependencies during quantization. Our approach optimizes multiple transformer blocks within a sliding window with overlapping, allowing for more effective and non-local optimization of quantization parameters. Using the CBD method, CBQ incorporates a LoRA-Rounding technique, employing two low-rank matrices to learn adaptive compensation values for quantized weights. Notably, we jointly optimize the compensation matrices and the step sizes of weights and activations within the overlapping window, which helps manage intra-layer dependencies to rectify weight quantization errors while preserving training efficiency. Furthermore, CBQ introduces a novel unified coarse-to-fine pre-processing (CFP) strategy from a statistical perspective to evaluate outliers in weights and activations, precisely handling outliers while minimizing damage to normal channels. CFP employs a quartile criterion to initially estimate the range of outliers and then assesses the intra-class and inter-class distances between outliers and normal values to precisely identify their locations. This approach facilitates the truncation of weight outliers and the application of equivalent scaling to activation outliers.

The contributions of this paper are summarized as follows:

- We performed a comprehensive analysis of the error sources in low-bit quantization scenarios for LLMs, and theoretically demonstrated the significant impact of intra-layer and inter-layer dependencies on the effectiveness of model quantization.
- We propose CBQ, a unified PTQ method designed for LLMs, incorporating a cross-block reconstruction strategy that introduces a Cross-Block Dependency (CBD) mechanism to preserve the model's internal dependencies during quantization, and LoRA-Rounding to utilize intra-layer dependencies for optimizing adaptive compensation matrices.
- We design a coarse-to-fine pre-processing strategy (CFP) that can simultaneously detect and manage outliers in both weights and activations, effectively preventing disruption to normal activation channels and weights.
- Extensive experiments demonstrate the effectiveness of our method in ultra-low bit quantization settings such as W4A4, W4A8, and W2A16. Notably, it outperforms state-of-the-art methods across diverse models and benchmark datasets.

## 2 MOTIVATION

To analyze the sources of quantization errors in large models when quantizing weights or activations, we assume a matrix $M$ representing a set of weights or activations as the current quantization target, and $\mathcal{L}$ denotes the quantization loss of the model under this matrix. Let $\varepsilon$ denote a small perturbation introduced by quantization and $\mathcal{L}(M)$ represent the task loss that we aim to minimize. Then, we can derive the following equation within the Taylor expansion:

$$\mathbb{E}[\mathcal{L}(M+\varepsilon) - \mathcal{L}(M)] \approx \mathbb{E}[\varepsilon^T \cdot \frac{\partial \mathcal{L}}{\partial M} + \frac{1}{2}\varepsilon^T \frac{\partial^2 \mathcal{L}}{\partial M^2}\varepsilon + O(||\varepsilon||^3)] \approx \varepsilon^T \cdot g^{(M)} + \frac{1}{2}\varepsilon^T \cdot \mathbf{H}^{(M)} \cdot \varepsilon \quad (1)$$

As discussed in previous work (Frantar et al. (2022b)), The error $\varepsilon$ introduced by quantization is sufficiently small, the higher-order terms in the Taylor expansion can be neglected. Therefore, we analyze the first- and second-order terms, $g^{(M)}$ and $\mathbf{H}^{(M)}$, which can be defined as follows.

$$g^{(M)} = \mathbb{E}[\nabla_M \mathcal{L}(M)] = \sum_i^K \frac{\partial \mathcal{L}}{\partial M_i} \quad (2)$$

$$\mathbf{H}^{(M)} = \mathbb{E}[\nabla_M^2 \mathcal{L}(M)] = \sum_i^K \sum_j^K \frac{\partial^2 \mathcal{L}}{\partial M_i \partial M_j} \quad (3)$$

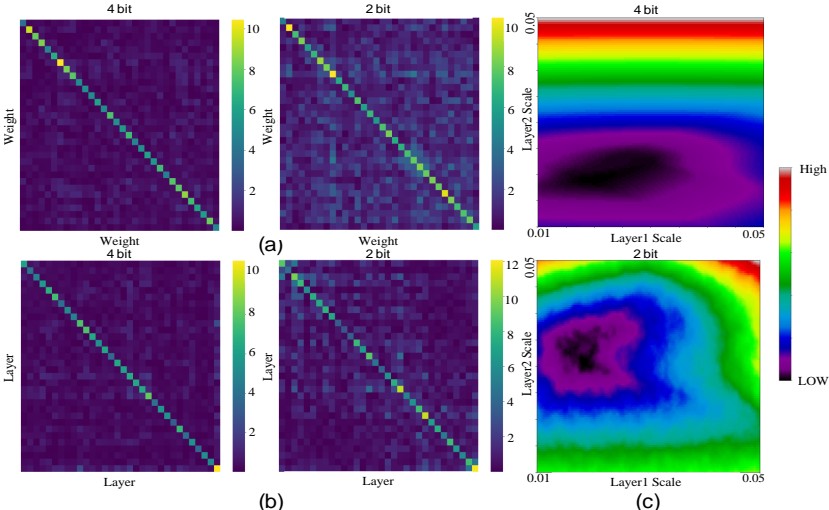

Figure 1: (a) Visualization of the absolute values of the Hessian matrix for weights within a single layer of LLAMA-7B, (b) Hessian matrix visualization of the loss with respect to the scale across 32 layers of LLAMA-7B, and (c) the relationship between the average scale of the first two transformer blocks in LLAMA-7B and the corresponding loss.

Let $K$ denote the number of elements in the LLM involved in the quantization. Using Equation 2 and 3, the influence of any two elements $i$ and $j$ on the final quantization loss can be calculated. From equations 1, 2, 3, it can be observed that when the quantization perturbation $\varepsilon$ is small, $||\varepsilon||^2$ is also small, allowing us to disregard the implications of the equation 3. In this case, the quantization error is primarily related to the current quantization target $M$, analogous to high-bit quantization. However, when performing low-bit quantization, $||\varepsilon||^2$ increases, necessitating consideration of the impact described by Equation 3. This indicates that when $i \neq j$, relationships between different $M$ are introduced. This relationship manifests in two aspects: when quantizing a single layer, it reflects intra-layer dependencies among parameters, and when quantizing the entire model, inter-layer dependencies must also be considered. Furthermore, given that the complexity of the Hessian matrix $\mathbf{H}$ is proportional to $\mathcal{O}(n^2)$, where $n$ represents the number of parameters, the growth in model size, both in terms of parameters and layers, leads to a marked intensification of intra-layer and inter-layer dependencies.

To better illustrate intra-layer and inter-layer dependencies, we visualize Equation 3 for both individual layers and the entire model using LLAMA-7B. Additionally, we present visualizations of the dependencies between adjacent blocks, as referenced in Figure 1.

By analyzing Figure 1, we observe a notable increase in the values of off-diagonal elements during lower-bit quantization. This increase indicates a strengthening of both inter-layer and intra-layer dependencies, with closer elements exhibiting stronger correlations. Furthermore, comparisons of the scales between adjacent layers provide a clearer understanding of the substantial impact that inter-layer dependencies have on final quantization outcomes in low-bit scenarios.

Therefore, taking into account both intra-layer and inter-layer dependencies, we present the quantization framework for LLMs under low-bit settings, which can be expressed by the following equation:

$$\arg \min_{h \subseteq \mathbf{H}_+} \sum_{k \in h} \mathbb{E}(T_k(W^k, X^k), QT_k(Q(W^k) + \Delta_W^k, Q(X^k))), \tag{4}$$

where $T$ and $QT$ represent the floating-point and quantized transformer blocks, respectively. $Q.(\cdot)$ represents the quantization process. $\mathbb{E}(\cdot)$ represents the metric to evaluate the reconstruction errors between outputs of quantized block and full-precision block. We jointly optimize all transformer blocks with inter-layer dependencies while compensating for intra-layer relationships using $\{\Delta_W^k | k \subseteq \mathbf{H}_+\}$.

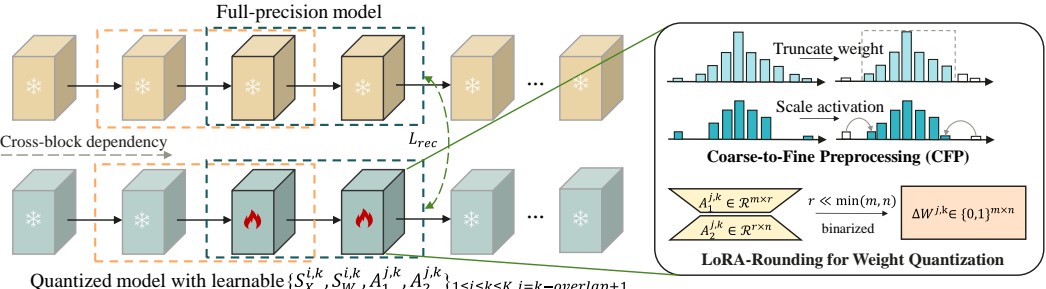

Figure 2: Workflow of the proposed CBQ. CBQ firstly utilizes a coarse-to-fine preprocessing to handle the outliers of weights and activations, and then employs a cross-block optimization strategy to learn quantization step sizes and weight adaptive rounding matrices with supervision from the corresponding full-precision model. This sequential block-wise method minimizes aggregate error propagation through cross-block dependency modeling.

## 3 METHOD

In this section, we introduce the proposed cross-block quantization framework tailored to LLMs. As illustrated in Fig. 2, CBQ firstly handles the outliers of weights and activations, and then jointly learns step sizes of weights and activations and weight-compensation matrices in a cross-block manner. CBQ reconstructs the output feature of the last block in each sliding window based on the corresponding supervision of the full-precision model.

### 3.1 CROSS-BLOCK RECONSTRUCTION

To maintain inter-layer dependencies, it is necessary to optimize the layers with significant dependencies together. As shown in Figure 1, the strongest dependencies are typically observed between adjacent layers. Therefore, we introduce a cross-block dependency (CBD) scheme using a sliding window approach. This scheme enables the simultaneous optimization of multiple blocks within the window. Furthermore, the two adjacent sliding windows have overlapping blocks, ensuring that the blocks between the windows are also interconnected. The CBD scheme enhances the connectivity and cooperation between blocks, enabling them to jointly contribute to the quantization process. This holistic optimization strategy leads to better overall performance and addresses the limitations of block-wise reconstruction in capturing cross-block dependencies. We formulate the optimization with the CBD scheme as

$$\underset{S_X^{i,k},S_W^{i,k},\Delta_W^{i,k}}{\arg\min} \mathbb{E}(T_{i,k}(W^{i,k},X^{i,k}), T_{i,k}(Q(W^{i,k}),Q(X^{i,k})), \tag{5}$$

where $1 \le i \le k \le K$, $T_{i,k}$ represents the blocks from block $i$ to block $k$ within one sliding window, and the same applies to the symbols $S_X^{i,k}$, $S_W^{i,k}$ and $\Delta_W^{i,k}$. The optimization object $\mathcal{L}_{rec}$ is as follow:

$$\mathcal{L}_{rec} = \mathbb{E}(T_{i,k}(W^{i,k},X^{i,k}), T_{i,k}(Q(W^{i,k}),Q(X^{i,k})) \tag{6}$$

For the distance metric, we incorporate $\mathcal{L}_2$ and Kullback-Leibler divergence (KLD) loss (Kullback & Leibler (1951)) to measure reconstruction error. KLD computes the likelihood distribution between output features that undergo the softmax function. It tends to suppress outliers in the feature space and enhance the robustness of the optimization process. By incorporating both terms, our method captures both the spatial distance and the distribution discrepancy, leading to a more comprehensive and robust optimization process. Then the distance metric is formulated as:

$$\mathbb{E}(h_1, h_2) = ||h_1 - h_2||_2 + D_{KL}(\sigma(h_1), \sigma(h_2)), \tag{7}$$

where $h_1$ and $h_2$ are hidden states from the outputs of full-precision blocks and quantized blocks, respectively. $\sigma$ is the softmax function. $||\cdot||_2$ represents the $\mathcal{L}_2$ distance and $D_{KL}(\cdot)$ represents the KLD distance. We provide the ablation study on the loss functions in Appendix B.Table 5.

## 3.2 LoRA-Rounding for weight quantization

AdaRound (Nagel et al. (2020)) introduces to learn a better weight-rounding matrix for post-training quantization that adapts to the data and the task loss. As shown in Eq. 8, we can obtain the weight-rounding matrix $\Delta_W \in \mathbb{R}^{d \times k}$ with a learnable matrix $V \in \mathbb{R}^{d \times k}$ with a rectified sigmoid function:

$$\Delta_W = \text{Clip}(\text{Sigmoid}(V)(\zeta - \gamma) + \gamma, 0, 1), \tag{8}$$

where $\zeta$ and $\gamma$ are stretch parameters and are fixed to 1.1 and -0.1, and $\text{Clip}(\cdot)$ clamps the inputs into a given range. The size of the weight-rounding matrix $\Delta_W$ is the same as the original weights.

When the transformer blocks are within the overlap of the CBD sliding window mechanism, the rounding matrix can serve as an effective representation of intra-layer dependencies. We utilize it as a compensation matrix and jointly optimize it with the quantization step sizes for weights and activations, which can be expressed as follows:

$$\underset{S_X^{i,k}, S_W^{i,k}, \Delta_W^{j,k}}{\arg\min} \quad \mathbb{E}(T_{i,k}(W^{i,k}, X^{i,k}), T_{i,k}(Q(W^{i,k}) + \Delta_W^{j,k}, Q(X^{i,k}))) \tag{9}$$

$$\text{s.t. } j = k + 1 - overlap \tag{10}$$

However, as shown in Experiment Table 3b, we found that LLMs with billion-level parameters result in an exceptionally large $\Delta_W^{j,k}$, which can lead to significant computational overhead and substantially impact the convergence of training. (Shao et al. (2023)) has also mentioned that AdaRound cannot be applied to models with billions of parameters due to the vast solution space, which aligns with our experimental findings. Thus, we employ low-rank adaptive learning on the compensation matrices, decomposing $V$ with much smaller low-rank matrices, and only optimize them in post-training quantization, the decomposition is defined as:

$$\Delta_W = A_1 \times A_2, A_1 \in \mathbb{R}^{d \times r}, A_2 \in \mathbb{R}^{r \times k}, \tag{11}$$

Where the rank $r << \min(d, k)$, we utilize a random Gaussian initialization for $A_1$ and zero for $A_2$, thus $\Delta_W$ is set to zero at the beginning of post-training quantization. During training, each element of $\Delta_W$ is encouraged into 0 or 1 with a regularizer loss:

$$\mathcal{L}_{com} = \sum_{i,j} 1 - |2\Delta_W(i,j) - 1|^\beta, \tag{12}$$

Where $\beta$ is a annealing factor. Following (Nagel et al. (2020)), $\beta$ is set to higher in the initial phase and set to lower in the later phase of the optimization to encourage it to converge to 0 or 1. We also conduct $\Delta_W = \lfloor \Delta_W \rceil$ in the later phase of the optimization to force each element into $\{0, 1\}$ exactly.

**Compared with vanilla AdaRound for LLMs**. The proposed LoRA-Rounding reduces the number of learnable parameters from $d \times k$ to $(d + k) \times r$ and changes the training strategy, significantly accelerating the optimization process, we conduct ablation experiments in the next section 5.3.

## 3.3 Overall loss

In summary, by leveraging CBD and a low-rank decomposition of the weight-compensated matrix, We slide the window to the last block with an interval and update all the quantization parameters $S_W, S_X, A_1, A_2$ within the window, ensuring the preservation of both intra-layer and inter-layer relationships of the model, thereby achieving optimal performance. The total loss for optimizing the $i^{th}$ block to the $k^{th}$ block within a sliding window is formulated as

$$\mathcal{L}_{total} = \mathcal{L}_{rec} + \gamma \mathcal{L}_{com}, \tag{13}$$

where the $\gamma$ is the hyper-parameter to balance the reconstruction error and compensation error.

## 3.4 Coarse-to-fine pre-processing

Outlier handling is crucial in quantizing LLMs. Figure 3 in Appendix F illustrates the prevalent outliers in weights and activations, which pose significant challenges to the quantization process. Although there are many existing studies based on outliers problem, these studies typically focus on outliers in either weights or activations individually, such as in (Chee et al. (2024); Wei et al. (2022b);

Xiao et al. (2022)). However, there is no precise strategy that can simultaneously detect and handle outliers in both weights and activations. This single-mode approach can potentially damage normal activation channels and weights due to incorrect outlier detection. To address this issue, we discard the previous assumption of normal distributions for weights and activations (Wu et al. (2023)), and based on statistical principles (Massart et al. (2005)), propose a coarse-to-fine pre-processing strategy to decouple the outlier handling in activations and weights. Relevant theoretical details can be found in Appendix F.

The comprehensive algorithm of the outlier detection is illustrated in Algorithm 1 in Appendix K, which is divided into two stages.

**Coarse-grained detection.** In the first stage, we perform coarse-grained detection by calculating the lower and upper quartile values ($Q_1$ and $Q_3$) and the interquartile range ($IQR$) (Massart et al. (2005)) in the numerical distribution (either activations or weights). Based on these calculations, we obtain a coarse outlier set $O = \{x | x > T, x \in X\}$, where $T = Q_3 + \lambda_1 IQR$ and $\lambda_1$ is set to 1.5. This stage greatly reduces the search space for outlier detection.

**Fine-grained detection.** In the second stage, we perform fine-grained detection by searching for a threshold that splits the coarse outlier set into an outlier subset $O_{outlier}$ and a reserved subset $O_{reserved}$. The goal is to minimize the intra-set variance $M_{intra} = \text{Var}(O_{reserved})$ while maximizing the distance between the two subsets $M_{inter} = (\text{Min}(O_{outlier}) - \text{Max}(O_{reserved}))^2$. To balance these objectives, we define a metric $M = M_{inter} - \lambda_2 M_{intra}$, where $\lambda_2 = 1.0$. By minimizing this metric, we can effectively identify outliers and distinguish them from the remaining data.

Removing outliers in weights has minimal impact on performance, whereas outliers in activations, particularly in specific channels, can greatly affect performance if directly removed. Consequently, our approach involves truncating weight outliers and scaling outliers in activations based on the detected outliers in both weights and activations. Figure 3 in Appendix F provides visual evidence of weight outliers being truncated within the outlier group.

The scaling factor $s_i$ for the activation tensor in $i^{th}$ channel (represented as $X_i$) is determined by the maximum absolute value of the truncated outlier set $O^*$:

$$s_i = \sqrt{\text{Max}(|X_i|)/\text{Max}(O^*)}. \tag{14}$$

This scaling factor is then applied to update weights and activations following prior work (Wei et al. (2023)) to counteract destabilizing fluctuations from remaining outliers.

## 4 RELATED WORK

**Post-training quantization.** The post-training quantization (PTQ) algorithm (Nagel et al. (2021); Wu et al. (2020; 2023); Zhang et al. (2018)) converts the pre-trained full-precision network into a fixed-point network with a few unlabeled calibration data and computational overhead, which enables fast deployment on various devices. Recent post-training quantization methods have been widely explored in vision models (Liu et al. (2021); Hubara et al. (2021); Frantar & Alistarh (2022); Cai et al. (2020); Li et al. (2022)). Some techniques like AdaQuant (Hubara et al. (2020)), AdaRound (Nagel et al. (2020)), and BRECQ (Li et al. (2021)) minimize the distance between floating point and quantized model outputs to optimize quantization parameters. While BRECQ incorporates Fisher information and jointly optimizes layers within each residual block, it still obtains sub-optimal performance for not capturing interactions across neighbouring residual blocks. The proposed CBQ improves quantization accuracy that accounts for dependencies between adjacent blocks.

**Quantization for large language models.** Existing large language models such as BLOOM (Laurençon et al. (2022)), OPT (Zhang et al. (2022)), and LLAMA (Touvron et al.; 2023)) contain tens of billions of parameters, and require massive memory footprint and computation requirements in the inference (Ashkboos et al. (2023); Wang et al. (2010); Bolya & Hoffman (2023); Brown et al. (2020a); Jacob et al. (2018)). Recent works have been proposed to compress LLMs with post-training quantization methods that do not require a complete training procedure and access to a full training dataset. LLM.int8() (Dettmers et al.), ZeroQuant (Yao et al. (2022)) and nuQmm (Park et al. (2022)) focus on quantizing the parameters with mixed-precision decomposition

scheme, representing the outliers with 16-bit and others with 8-bit. These methods can not truly accelerate the inference of LLMs for that is hard to implement on hardware. Other methods like GPTQ (Frantar et al. (2022b)) and AWQ (Lin et al. (2023)) can efficiently quantize LLMs but they focus on FP16 activations and INT4 weights, which can not benefit from the integer matrix multiplication of existing AI accelerators. Additionally, Some methods like SmoothQuant (Xiao et al. (2022)), Outlier Suppression (Wei et al. (2022b)), Outlier Suppression+ (Wei et al. (2023)) and QLLM (Liu et al. (2023a)) aim at processing activation outliers (Zhao et al. (2019)) and lack optimization for the weight quantization. Moreover, these methods rely on hand-craft quantization strategies which are tuned based on extensive experimentation for optimization. Recent block reconstruction-based PTQ method OmniQuant (Shao et al. (2023)),QLLM (Liu et al. (2023a)), have experienced significant accuracy degradation in low-bit settings. In contrast, CBQ introduces a more precise outlier detection strategy and optimizes the reconstruction process through CBD and LoRA-Rounding mechanisms by maintaining both intra-layer and inter-layer dependencies.

## 5 EXPERIMENTS

### 5.1 SETUP

**Models and datasets.** We conduct experiments on large language models with different sizes, including OPT (Zhang et al. (2022)) and LLAMA (Touvron et al.) families. We validate our quantization scheme on various datasets which are divided into two categories. One is reported by the perplexity metric of language generation experiments on C4 (Raffel et al. (2020)) and WikiText2 (Merity et al. (2016)). The other is reported by the accuracy metric of zero-shot language tasks (Gao et al. (2021)) on PIQA (Bisk et al. (2020a)), HellaSwag (Clark et al. (2018)), ARC (Clark et al. (2018)), Mutual (Cui et al. (2020)) and Ehics (Hendrycks et al. (2020a)).

**Quantization setting.** To thoroughly evaluate performance, we test extensive quantization schemes including weight-only quantization down to W4A16 and W2A16, as well as joint weight-activation quantization for ultra-low bitwidths like W4A8, and W4A4. This extensive assessment across varying bitwidths provides a robust analysis of our proposed method. Also, In alignment with prior research (Shao et al. (2023); Liu et al. (2023a); Frantar et al. (2022c)), we use per-channel weight quantization and per-token activation quantization.

**Baseline methods.** For weight-only quantization settings, we selected GPTQ (Frantar et al. (2022b)) as the baseline quantization method in our experiments. This represents the most prevalent technique for W4A16 quantization of language models. Furthermore, we compare our CBQ with OmniQuant (Shao et al. (2023)) and QLLM (Liu et al. (2023a)), which is the state-of-the-art method based on block reconstruction. We include a comparison of our CBQ method with the groupwise quantization method RPTQ (Yuan et al. (2023)), which is widely employed in the W4A8 setting.

**Implementation details.** Following the setting of previous work (Frantar et al. (2022b); Liu et al. (2023b); Yao et al. (2024); Yuan et al. (2023)), our calibration dataset comprises 128 randomly selected 2048-token segments from C4 to ensure standardized benchmarking. To balance quantization performance and training speed, we utilize sliding windows containing two blocks with 3 epochs per window. For the LoRA-Rounding technique, we set the rank $r$ to 5. The optimization process involves adjusting the learnable quantization step sizes ($S_X$ and $S_W$) and the weight-rounding matrix ($\delta_W$) with learning rates of $1e-4$, $1e-3$, and $1e-4$, respectively. To manage the learning rate, we utilize the CosineAnnealingLR scheduler. We quantize all models using a mini-batch size of 1 on a single GPU. This configuration allows efficient cross-block dependency modeling while sufficiently propagating information across windows.

### 5.2 EXPERIMENTAL RESULTS

**Evaluation on zero-shot datasets with accuracy.** Results on multiple zero-shot benchmarks using accuracy as the evaluation metric demonstrate CBQ's capabilities on LLMs including OPT (30B, 66B) and LLAMA (30B, 65B) (as shown in Table 1). Across almost all datasets, CBQ outperforms existing quantization methods by over 2% and reduces the accuracy gap with the full precision model to within 1% under the W4A16, W2A16 and W4A8 quantization settings. This demonstrates

Table 1: Evaluation on multiple zero-shot datasets with the accuracy ↑ metric, where the Mutual dataset is evaluated with the Mean Reciprocal Rank/Recall@1/Recall@2 metrics. CBQ* represents that the experiments conducted with 2-bit weight-only quantization did not fully quantize the model but only the FC2 layer of the first and last transformer blocks are converted to 4-bit precision.

| Models | #Bits | Methods | PIQA | HellaSwag | ARC-C | ARC-E | Mutual | Ethics |
|---|---|---|---|---|---|---|---|---|
| OPT-30B | FP | - | 78.18 | 72.27 | 38.14 | 65.40 | 69.72 / 48.83 / 74.98 | 60.28 |
| | W4A16 | GPTQ | 78.10 | 71.50 | 37.54 | 63.88 | 68.64 / 47.40 / 74.27 | 58.64 |
| | | OmniQuant | 78.06 | 71.29 | 37.98 | 65.19 | 69.34 / 48.64 /74.71 | 58.73 |
| | | CBQ | 78.36 | 72.23 | 38.06 | 65.35 | 69.77 / 49.32 / 74.47 | 61.31 |
| | W2A16 | GPTQ | 66.38 | 52.55 | 28.41 | 43.86 | 64.50 / 41.08 / 68.62 | 52.15 |
| | | OmniQuant | 72.85 | 66.81 | 35.98 | 56.65 | 62.36 / 43.12 / 68.62 | 53.64 |
| | | CBQ | 76.19 | 66.90 | 36.23 | 59.72 | 68.20 / 47.29 / 72.23 | 52.10 |
| | | CBQ* | 78.29 | 71.18 | 36.95 | 64.01 | 69.49 / 48.76 / 75.06 | 60.05 |
| | W4A8 | OmniQuant | 77.20 | 71.17 | 37.11 | 64.60 | 68.81 / 47.51 / 74.60 | 59.17 |
| | | RPTQ | 76.93 | 71.25 | 37.45 | 63.46 | 68.98 / 47.67 / 74.75 | 59.21 |
| | | CBQ | 78.26 | 71.55 | 37.89 | 64.92 | 69.01 / 47.72 / 74.81 | 59.23 |
| | W4A4 | OmniQuant | 75.38 | 67.47 | 33.27 | 61.23 | 67.12 / 45.14 / 72.34 | 56.30 |
| | | CBQ | 75.89 | 67.49 | 34.81 | 61.58 | 67.73 / 45.94 / 73.14 | 56.60 |
| OPT-66B | FP | - | 79.81 | 74.86 | 40.01 | 67.26 | 69.84 / 48.87 / 74.94 | 58.14 |
| | W4A16 | GPTQ | 79.32 | 73.15 | 38.95 | 65.45 | 69.10/ 48.46 / 74.26 | 54.90 |
| | | OmniQuant | 79.43 | 73.27 | 38.97 | 66.85 | 69.04 / 48.45 / 74.24 | 55.87 |
| | | CBQ | 79.71 | 74.69 | 39.18 | 67.38 | 69.50 / 48.65 / 74.83 | 57.35 |
| | W2A16 | GPTQ | 54.24 | 52.55 | 23.04 | 32.28 | 60.45 / 35.56 / 61.74 | 49.50 |
| | | OmniQuant | 77.01 | 73.10 | 34.65 | 66.32 | 65.26 / 43.23 / 70.47 | 51.46 |
| | | CBQ | 78.05 | 73.45 | 35.37 | 66.84 | 67.34 / 45.31 / 72.45 | 55.95 |
| | | CBQ* | 79.21 | 74.32 | 38.96 | 67.11 | 69.32 / 48.35 / 74.69 | 56.78 |
| | W4A8 | OmniQuant | 77.12 | 73.56 | 37.65 | 65.89 | 68.25 / 47.63 / 73.85 | 56.93 |
| | | RPTQ | 77.52 | 74.01 | 38.82 | 64.60 | 68.54 / 47.87 / 73.94 | 56.95 |
| | | CBQ | 79.12 | 74.21 | 39.25 | 67.16 | 69.07 / 48.32 / 74.53 | 56.98 |
| | W4A4 | OmniQuant | 77.85 | 71.76 | 37.20 | 63.29 | 68.20 / 46.61 / 73.02 | 55.54 |
| | | CBQ | 78.01 | 72.34 | 37.56 | 63.78 | 68.76 / 47.20 / 73.56 | 55.82 |
| LLAMA1-30B | FP | - | 80.09 | 79.21 | 45.39 | 58.92 | 72.45 / 53.49 /78.21 | 57.42 |
| | W4A16 | GPTQ | 79.62 | 78.81 | 44.54 | 58.42 | 72.30 / 52.93 / 77.44 | 56.30 |
| | | OmniQuant | 79.83 | 78.95 | 46.26 | 59.34 | 72.29 / 53.38 / 77.65 | 56.21 |
| | | CBQ | 80.12 | 79.11 | 46.65 | 59.89 | 72.85 / 53.95 / 78.56 | 57.85 |
| | W2A16 | GPTQ | 51.03 | 26.34 | 26.02 | 28.87 | 56.53 / 29.80 / 58.13 | 52.72 |
| | | OmniQuant | 77.23 | 73.85 | 43.52 | 55.23 | 70.62 / 50.89 / 74.96 | 50.36 |
| | | CBQ | 77.23 | 75.05 | 42.93 | 57.12 | 69.96 / 49.93 /75.65 | 56.35 |
| | | CBQ* | 80.09 | 78.85 | 45.05 | 58.42 | 72.74 / 53.95 /78.44 | 57.65 |
| | W4A8 | OmniQuant | 78.95 | 76.34 | 44.62 | 57.36 | 71.03 / 52.89 / 77.06 | 57.05 |
| | | CBQ | 79.34 | 78.98 | 45.13 | 58.45 | 71.35 / 53.23 / 77.64 | 57.19 |
| | W4A4 | OmniQuant | 71.21 | 64.65 | 34.47 | 49.45 | 67.10 / 45.37 / 71.44 | 47.69 |
| | | QLLM | 73.83 | 67.91 | 38.40 | 50.67 | - | - |
| | | CBQ | 76.33 | 72.74 | 42.92 | 54.50 | 70.12 / 50.45 / 74.73 | 48.70 |
| LLAMA1-65B | FP | - | 80.79 | 80.72 | 46.24 | 58.71 | 73.03 / 54.17 / 79.12 | 61.75 |
| | W4A16 | GPTQ | 80.79 | 79.86 | 45.45 | 58.13 | 72.89 / 53.84 / 78.57 | 58.45 |
| | | OmniQuant | 81.01 | 80.30 | 45.74 | 58.41 | 72.99 / 54.06 / 79.11 | 60.12 |
| | | CBQ | 81.12 | 80.76 | 45.98 | 58.64 | 73.06 / 54.29 / 78.89 | 61.49 |
| | W2A16 | GPTQ | 56.47 | 33.31 | 25.43 | 31.69 | 59.28 / 33.86 / 60.49 | 50.93 |
| | | OmniQuant | 79.50 | 72.38 | 40.35 | 52.56 | 69.50 / 48.64 / 74.94 | 52.64 |
| | | CBQ | 78.12 | 74.28 | 41.64 | 55.35 | 70.67 / 50.80 / 75.51 | 55.95 |
| | | CBQ* | 81.07 | 80.51 | 45.81 | 57.45 | 73.43 / 54.96 / 79.23 | 61.35 |
| | W4A8 | OmniQuant | 79.21 | 78.96 | 44.63 | 57.68 | 72.24 / 53.89 / 78.65 | 59.68 |
| | | CBQ | 79.95 | 79.30 | 45.43 | 58.12 | 72.83 / 54.27 / 79.02 | 61.25 |
| | W4A4 | OmniQuant | 71.81 | 66.81 | 35.92 | 48.02 | 68.49 / 47.29 / 73.70 | 57.19 |
| | | QLLM | 73.56 | 70.94 | 39.68 | 52.06 | - | - |
| | | CBQ | 77.69 | 76.65 | 43.25 | 56.01 | 70.93 / 51.35 / 75.62 | 57.50 |

stronger zero-shot capability. Moreover, unlike current techniques, CBQ uniquely achieves ultra-low quantization down to W4A4 while maintaining a higher performance than the state-of-the-arts. The consistent gains verify the generalization of CBQ's innovations across models and datasets.

**Evaluation on generation datasets with perplexity.** Results in Table 2 demonstrate our method's generation performance on C4, WikiText2 using weight-only quantized OPT and LLAMA models. Focusing on ultra-low bitwidths, we achieve over 1% higher perplexity versus GPTQ at W4A16. These consistent improvements at low bitwidths highlight our advantages in preserving generative quality under aggressive compression rates.

Table 2: Evaluation quantization on generation datasets with the perplexity (PPL) ↓ metric, where 'OmniQ' represents OmniQuant.

| #Bits | Methods | OPT-30B | | OPT-66B | | LLAMA1-30B | | LLAMA1-65B | |
|-------|---------|---------|------|---------|------|------------|------|------------|------|
| | | C4 | Wiki | C4 | Wiki | C4 | Wiki | C4 | Wiki |
| FP | - | 10.69 | 9.56 | 10.28 | 9.34 | 5.98 | 4.10 | 5.62 | 3.53 |
| W4A16 | GPTQ | 10.80 | 9.63 | 10.50 | 9.55 | 6.16 | 4.34 | 5.77 | 3.77 |
| | OmniQ | 10.80 | 9.71 | 10.63 | 9.37 | 6.06 | 4.19 | 5.68 | 3.62 |
| | CBQ | **10.73** | **9.65** | **10.31** | **9.41** | **6.03** | **4.14** | **5.62** | **3.59** |
| W2A16 | GPTQ | 1.6e4 | 9.1e3 | 4.3e3 | 6.3e3 | 7.2e3 | 1.3e4 | 8.8e3 | 1.1e4 |
| | OmniQ | 12.80 | 11.00 | 12.13 | 10.59 | 9.02 | 7.14 | 7.78 | 6.01 |
| | CBQ | **12.01** | **10.51** | **11.19** | **10.25** | **7.65** | **5.58** | **7.42** | **5.25** |
| | CBQ* | **10.92** | **10.26** | **10.39** | **9.48** | **6.02** | **4.21** | **5.73** | **3.73** |
| W4A8 | OmniQ | 10.96 | 9.95 | 10.73 | 9.52 | 6.45 | 4.58 | 6.12 | 3.96 |
| | RPTQ | 11.01 | 10.22 | 10.57 | 9.46 | - | - | - | - |
| | CBQ | **10.86** | **9.83** | **10.42** | **9.44** | **6.25** | **4.32** | **5.96** | **3.84** |
| W4A4 | OmniQ | 11.89 | 10.60 | 11.35 | 10.29 | 12.49 | 10.33 | 11.28 | 9.17 |
| | QLLM | - | - | - | - | 11.51 | 8.37 | 8.89 | 6.87 |
| | CBQ | **11.79** | **10.34** | **11.02** | **9.45** | **9.73** | **7.96** | **7.52** | **5.89** |

## 5.3 ABLATION STUDY

To analyze the contribution of each component in our proposed CBQ method, we performed ablation experiments on the LLAMA-7B model under W4A4.

Table 3: Ablation studies on the proposed CBD, CFP and LoRA-Rounding.

(a) Ablation of the CFP

| Method | C4 ↓ | Wiki ↓ |
|--------|------|--------|
| w/o outlier pre-processing | 1082.68 | 1128.33 |
| w/ OMSE (Choukroun et al. (2019)) | 76.43 | 47.81 |
| w/ Percentile (Zhou et al. (2017)) | 71.62 | 45.86 |
| w/ OS (Wei et al. (2022b)) | 41.57 | 26.36 |
| w/ Smoothquant (Xiao et al. (2022)) | 33.21 | 25.26 |
| w/ CFP-Activation | 23.48 | 19.75 |
| w/ CFP-Weight + CFP-Activation | **21.98** | **17.95** |
| w/ OMSE + CBQ-Recon. | 25.34 | 19.53 |
| w/ Percentile + CBQ-Recon. | 25.62 | 19.45 |
| w/ OS + CBQ-Recon. | 17.83 | 13.89 |
| w/ Smoothquant + CBQ-Recon. | 15.69 | 12.24 |
| w/ CFP-Weight+Act + CBQ-Recon. | **13.29** | **10.63** |

(b) Ablation of the LoRA-Rounding

| Method | PPL ↓ | | | |
|--------|------|------|---------|----------|
| | C4 | Wiki | #Epochs | GPU (GB) |
| w/o Rounding | 14.32 | 11.46 | 3 | 18.83 |
| w/ Adarounding | 14.56 | 11.64 | 3 | 27.73 |
| w/ Rounding | 13.86 | 10.98 | 3 | 27.73 |
| w/ Rounding | 13.58 | 10.72 | 6 | 27.73 |
| w/ LoRA-Rounding | **13.29** | **10.63** | 3 | 21.01 |

(c) Ablation on the CBD

| #Num of blocks | Overlap | C4 ↓ | Wiki ↓ | GPU (GB) |
|----------------|---------|-------|--------|----------|
| 1 | 0 | 14.57 | 11.98 | 17.2 |
| 2 | 0 | 14.23 | 11.35 | 21 |
| | 1 | 13.29 | 10.63 | 21 |
| 4 | 0 | 14.32 | 11.45 | 39 |
| | 1 | 13.27 | 10.60 | 39 |
| | 2 | 12.56 | 9.56 | 39 |
| | 3 | **12.32** | **9.45** | 39 |

**Cross-block dependency.** To analyze the impact of our proposed CBD method, we performed ablation experiments in Table 3c. Results demonstrate performance gains as the number of blocks jointly processed per sliding window increases, validating CBD's ability to model inter-block dependencies. Furthermore, utilizing overlapping blocks between adjacent sliding windows supplements cross-window relational representation. This redundancy helps capture nuanced block interactions and enables additional accuracy improvements. Overall, these ablation studies highlight the benefits of CBD for progressively accumulating and propagating cross-block knowledge during CBQ optimization. For additional experimental results, please refer to Appendix D and E.

**LoRA-Rounding.** As shown in Table 3b, 'w/ Rounding' indicates a modification in the training strategy for the compensation matrix, compared to the traditional 'w/ Adarounding' approach. This change leads to a significant improvement in accuracy. Overall, LoRA-Rounding leverages low-rank decomposition to reduce the number of learnable parameters and adjusts the training strategy, which not only decreases GPU memory consumption but also enhances training speed.

**Coarse-to-fine pre-processing.** As shown in Table 3a and Table 10 in Appendix F, CFP demonstrates advantages in both weight-activation quantization and weight-only quantization. Furthermore, we conduct a reconstruction optimization process, referred to as 'CBQ-Recon.', based on the preprocessed weights and activations. This two-pronged pre-processing effectively reduces outliers which are not adequately handled by existing preprocessing techniques like OS (Wei et al. (2022b)), Smoothquant (Xiao et al. (2022)) *etc*.

## 6 CONCLUSION

In this work, we conduct a detailed analysis of error sources in LLMs under low-bit quantization and identify the critical role of intra-layer and inter-layer dependencies. To address these challenges, we propose CBQ, a novel method that employs a cross-block reconstruction strategy alongside Lora-Rounding compensation matrices. This approach effectively establishes long-range inter-layer dependencies while capturing comprehensive intra-layer dependencies, surpassing traditional layerwise and block-wise reconstruction techniques. Additionally, we introduce CFP, a technique designed to simultaneously detect and manage outliers in both weights and activations. Our experimental results demonstrate that CBQ significantly outperforms existing PTQ, achieving substantial improvements in ultra-low bit precision across a variety of tasks, while also offering enhanced computational efficiency by reducing training resource demands.

## ACKNOWLEDGEMENT

We gratefully acknowledge the support of MindSpore, CANN (Compute Architecture for Neural Networks) and Ascend AI Processor used for this research.

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

# A OVERVIEW

Table 4: Comparison of different quantization methods for LLMs.

| Method | Quantize W/A | Gradient-Based | Cross-Block Dependency | Weight Outlier | Activation Outlier | Rounding Error |
|---|---|---|---|---|---|---|
| GPTQ (Frantar et al. (2022b)) | ✓ / ✗ | ✗ | ✗ | ✗ | ✗ | ✗ |
| RPTQ (Yuan et al. (2023)) | ✓ / ✓ | ✗ | ✗ | ✗ | ✓ | ✗ |
| OS+ (Wei et al. (2023)) | ✓ / ✓ | ✗ | ✗ | ✗ | ✓ | ✗ |
| SmoothQuant (Xiao et al. (2022)) | ✓ / ✓ | ✗ | ✗ | ✗ | ✓ | ✗ |
| OmniQuant (Shao et al. (2023)) | ✓ / ✓ | ✓ | ✗ | ✓ | ✓ | ✗ |
| QLLM (Liu et al. (2023a)) | ✓ / ✓ | ✓ | ✗ | ✗ | ✓ | ✗ |
| CBQ (Ours) | ✓ / ✓ | ✓ | ✓ | ✓ | ✓ | ✓ |

In Table 4, we compare the designed components of our CBQ with the existing quantization methods for LLMs. We can observe a comparison of the different components incorporated in various quantization methods LLMs. Our proposed CBQ method stands out by including multiple essential components to address the challenges associated with LLM quantization.

Firstly, CBQ ensures that both weight and activation values are quantized to improve computational efficiency and reduce memory requirements. This aligns with the requirements of other methods such as RPTQ, OS+, and SmoothQuant, and is different from GPTQ. Additionally, CBQ incorporates a gradient-based optimization approach, allowing for efficient optimization during the quantization process. This component is also present in OmniQuant and QLLM, signifying its significance in achieving accurate quantization results. Furthermore, CBQ introduces the cross-block dependency (CBD) component, enabling the modeling of long-range dependencies between adjacent blocks. This ensures better information flow and integration across multiple blocks, surpassing the capabilities of other methods such as OmniQuant and QLLM. Moreover, CBQ addresses the presence of weight and activation outliers, which can significantly impact the quantization process. By effectively handling these outliers, CBQ surpasses the capabilities of OS+, SmoothQuant, OmniQuant, and QLLM, which either do not consider or only partially address this issue. Lastly, CBQ accounts for rounding errors, a critical aspect of quantization. By incorporating a rounding error reduction scheme, CBQ ensures more accurate and reliable quantization results. This component is absent in all other compared methods.

In summary, our CBQ method outperforms existing quantization methods for LLMs by incorporating a comprehensive set of components that collectively address the challenges associated with LLM quantization. These components work synergistically to enhance the precision, accuracy, and efficiency of the quantization process, making CBQ a promising approach for LLM quantization.

# B ABLATION ON THE LOSS FUNCTIONS

Table 5: Ablation study on block-wise reconstruction loss functions.

| KL loss | L2 loss | C4 | Wiki |
|---|---|---|---|
| ✗ | ✓ | 13.82 | 11.13 |
| ✓ | ✗ | 13.84 | 11.12 |
| ✓ | ✓ | **13.29** | **10.63** |

To determine optimal loss formulations, we evaluate reconstruction errors using L2 alone, KLD alone, and a combination of them in Table 5. Ablation results demonstrate superior performance from KLD over L2, with the combined loss achieving further gains. This highlights the benefits of KLD for matching full-precision block distributions during CBQ optimization. Fusing both losses enables jointly minimizing absolute and divergence-based errors to improve overall block-wise reconstruction. Our analysis verifies the advantage of blended L2 + KLD loss for robustly optimizing blocks as interdependent components.

## C  THE CAPABILITY OF CBQ ON THE LLAMA2-7B

To demonstrate the effectiveness of CBQ, we evaluate CBQ on the LLAMA2-7B model across various datasets and observed that it delivers excellent results.

Table 6: Evaluation on multiple zero-shot datasets and generation datasets on the LLAMA2-7B

| #Bits | Methods | PIQA | HellaSwag | ARC-C | ARC-E | Mutual | Ethics | c4 ↓ | Wiki ↓ |
|---|---|---|---|---|---|---|---|---|---|
| FP | - | 76.93 | 72.95 | 40.69 | 53.21 | 70.92/51.12/75.84 | 52.63 | 6.97 | 5.47 |
| W4A16 | OmniQuant | 77.14 | 71.86 | 40.18 | 53.70 | 70.00/50.46/74.74 | 53.10 | 7.12 | 5.58 |
|  | **CBQ** | **77.34** | **72.23** | **40.22** | **53.66** | **70.49/50.90/74.83** | **53.13** | **7.05** | **5.52** |
| W3A16 | OmniQuant | 75.91 | 70.95 | 38.71 | 51.89 | 69.12/48.33/72.65 | 52.58 | 7.75 | 6.03 |
|  | **CBQ** | **76.25** | **71.34** | **39.21** | **52.36** | **69.35/49.02/73.15** | **52.67** | **7.56** | **5.89** |
| W2A16 | OmniQuant | 68.71 | 53.43 | 30.88 | 39.81 | 65.12/42.21/69.18 | 50.54 | 12.72 | 9.62 |
|  | **CBQ** | **71.59** | **60.28** | **32.93** | **45.74** | **66.22/44.35/69.63** | **57.22** | **11.30** | **8.01** |
| W4A4 | QLLM | 67.68 | 58.45 | 30.89 | 44.4 | - | - | 13.26 | 11.75 |
|  | OmniQuant | 65.94 | 53.53 | 30.80 | 43.94 | 64.83/41.87/68.84 | 47.29 | 18.39 | 14.61 |
|  | **CBQ** | **68.25** | **57.34** | **31.56** | **46.23** | **64.89/41.87/68.74** | **47.59** | **12.56** | **11.32** |
| W4A8 | **CBQ** | **76.85** | **72.06** | **40.16** | **53.34** | **70.23/50.12/74.89** | **52.56** | **7.12** | **5.72** |
| W6A6 | OmniQuant | 76.82 | 72.13 | 39.33 | 53.36 | 69.57/49.67/73.62 | 52.62 | 7.48 | 5.87 |
|  | **CBQ** | **77.58** | **72.14** | **40.27** | **53.87** | **70.16/50.22/74.83** | **53.02** | **7.24** | **5.67** |

## D  THE POTENTIAL FOR SCALING WITH CBD

To further investigate the scaling potential of cross-block dependency (CBD), we conducted additional experiments to explore whether increasing its scale could lead to further performance improvements.

Table 7: The scaling capability of CBQ on the LLAMA-7B under W4A4

| #Num of blocks | Overlap | C4↓ | Wiki↓ |
|---|---|---|---|
| 1 | 0 | 14.57 | 11.98 |
| 2 | 0 | 14.23 | 11.35 |
|  | 1 | 13.29 | 10.63 |
| 4 | 0 | 14.32 | 11.45 |
|  | 1 | 13.27 | 10.60 |
|  | 2 | 12.56 | 9.56 |
|  | 3 | **12.32** | **9.45** |
| 8 | 0 | 13.56 | 10.78 |
|  | 4 | 11.91 | 9.01 |
|  | 7 | **11.86** | **8.96** |

Table 8: The capability of CBD on the LLAMA2-7B

| # Num of blocks | Overlap | LLAMA2-7b-W2A16 | | LLAMA2-7b-W4A4 | |
|---|---|---|---|---|---|
|  |  | C4 | Wiki | C4 | Wiki |
| 1 | 0 | 12.34 | 9.12 | 14.28 | 12.33 |
| 2 | 0 | 11.89 | 8.76 | 13.85 | 11.96 |
|  | 1 | 11.30 | 8.01 | 12.56 | 11.32 |
| 4 | 0 | 11.32 | 8.05 | 12.52 | 11.35 |
|  | 1 | 11.12 | 7.95 | 12.15 | 11.01 |
|  | 2 | 11.08 | 7.89 | 11.85 | 10.83 |
|  | 3 | 10.92 | 7.82 | 11.5 | 10.62 |

As shown in the Table 7,8, we validate the proposed cross-block quantization in the W2A16 and the W4A4 experimental settings. Our ablation analysis in these settings underscores the robustness and versatility of CBD, showcasing its inherent simplicity to ensure seamless deployment across various quantization settings.

# E    EFFICIENCY OF THE CBD

In order to further study the various performances with CBD, we performed the following experiments. This table below illustrates the training time, GPU memory usage, and the number of cross-block dependencies employed in the W2A16 quantization of the LLAMA-7B model.

Table 9: Ablation of the cross-block dependency (CBD) with W2A16.

| #Num of blocks | Overlap | C4↓ | Wiki ↓ | time (h) | GPU memory(GB) |
|---|---|---|---|---|---|
| 1 | 0 | 12.72 | 9.62 | 1.09 | 17.2 |
| 2 | 0 | 12.56 | 9.34 | 1.50 | 21 |
|   | 1 | 12.30 | 8.87 | 3.02 | 21 |
| 4 | 0 | 12.34 | 8.89 | 1.10 | 39 |
|   | 1 | 11.63 | 8.59 | 1.40 | 39 |
|   | 2 | 11.42 | 8.28 | 1.96 | 39 |
|   | 3 | **11.21** | **8.08** | 2.60 | 39 |

Our CBD considers dependencies between two blocks within a sliding window, distinguishing it from existing methods that focus solely on individual block dependencies. This unique design yields significant performance improvements while incurring additional GPU overhead. Additionally, by incorporating overlapping windows, CBD enhances cross-block dependencies without requiring additional GPU memory usage.

# F    THE THEORETICAL FOUNDATION OF CFP

Existing outlier detection methods often assume that data follow a normal distribution, which is not always strictly applicable to real-world datasets. Our approach avoids assuming specific data distributions, providing flexibility in capturing outliers across diverse datasets. The quartile criterion is robust to outliers, as it is not heavily influenced by extreme values.

We give two commonly used methods as follows:

- $3\sigma$ (sigma) rule: This method assumes that data follows a normal distribution. Typically, data points that are more than two to three standard deviations away from the mean are considered outliers.
- Percentile-based method: This method uses percentiles to detect outliers.

Our quartile criterion follows the existing analysis (Massart et al. (2005)), which includes the maximum value, minimum value, median, and upper and lower quartiles, to detect outliers. This approach does not require any assumptions about the distribution of the data and does not impose any restrictive requirements on the data. It simply portrays the true shape of the data, providing an objective way to identify outliers.

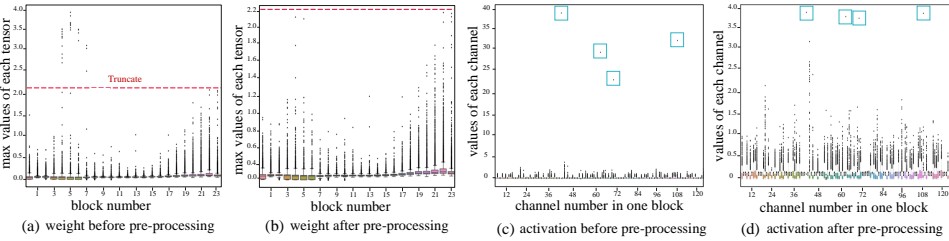

Figure 3: Outliers pre-processing for weights and activations. The red dashed line indicates the truncation threshold for weight outliers, and the deep blue line represents the reserved subset. The light blue boxes depict activation outliers that undergo per-channel scaling.

Table 10: The capability of CFP on the LLAMA2-7B

| Method | C4 ↓ | Wiki ↓ |
|---|---|---|
| w/o outlier pre-processing | 1082.68 | 1128.33 |
| w/ OMSE (Choukroun et al. (2019)) | 76.43 | 47.81 |
| w/ Percentile (Zhou et al. (2017)) | 71.62 | 45.86 |
| w/ OS (Wei et al. (2022b)) | 41.57 | 26.36 |
| w/ Smoothquant (Xiao et al. (2022)) | 33.21 | 25.26 |
| w/ CFP-Activation | 23.48 | 19.75 |
| w/ CFP-Weight + CFP-Activation | **21.98** | **17.95** |
| w/ OMSE + CBQ-Recon. | 25.34 | 19.53 |
| w/ Percentile + CBQ-Recon. | 25.62 | 19.45 |
| w/ OS + CBQ-Recon. | 17.83 | 13.89 |
| w/ Smoothquant + CBQ-Recon. | 15.69 | 12.24 |
| w/ CFP-Weight+Act + CBQ-Recon. | **13.29** | **10.63** |

## G    COMPARISON OF QUANTIZATION EFFICIENCY

Table 11: Comparison of training (GPU Hours) time of our CBQ with OmniQuant.

| LLAMA | 7B | 13B | 30B | 65B |
|---|---|---|---|---|
| OmniQuant | 1.1h | 2.2h | 4.5h | 8.9h |
| CBQ | 0.9h | 1.45 | 2.1h | 4.3h |

We evaluate the quantization efficiency of our weight-only CBQ quantization method and compare it to OmniQuant which is the representative reconstruction-based PTQ methods. The GPU training hours for both methods are shown in Table 11. The results demonstrate that the training cost of CBQ can be faster than OmniQuant, particularly for larger models. This indicates that our CBQ method offers an advantage in terms of training efficiency.

## H    ABLATION OF THE RANK OF LoRA-ROUNDING

The ablation of the rank of LoRA-Rounding is in the Table 12 below. It is observed that with lower ranks (3, 4, and 5), there is a slight improvement in performance. However, as the rank increases beyond 5, the performance starts to decline. Considering the limited training resources available for LLMs, it is worth noting that a larger rank in LoRA-Rounding results in a higher number of parameters that need to be optimized. This increased parameter complexity poses a significant challenge and ultimately leads to poorer performance and results.

Table 12: Ablation of the rank of LoRA-Rounding.

| Dataset | Rank = 3 | Rank = 4 | Rank = 5 | Rank = 6 | Rank = 7 |
|---|---|---|---|---|---|
| C4↓ | 13.4 | 13.35 | 13.29 | 13.46 | 13.98 |
| Wiki↓ | 10.89 | 10.71 | 10.63 | 10.86 | 11.05 |

## I    EVALUATION QUANTIZATION FOR A SERIES OF OPT MODELS

To further demonstrate the performance of the proposed CBQ, we conducted additional evaluations under the OPT models as shown in Table 13.

Table 13: Evaluation quantization for a series of OPT models on generation datasets with the perplexity ↓ metric

|  | #Bits | Methods | OPT-1.3B | OPT-2.7B | OPT-6.7B | OPT-13B |
|---|---|---|---|---|---|---|
| C4 | FP | - | 14.72 | 13.16 | 11.74 | 11.20 |
|  | W4A16 | GPTQ | 15.57 | 13.75 | 12.15 | 11.36 |
|  |  | CBQ | **15.42** | **13.56** | **11.92** | **11.29** |
|  | W2A16 | OmniQuant | 27.33 | 19.16 | 15.44 | 14.16 |
|  |  | CBQ | **15.99** | **13.83** | **12.19** | **11.52** |
| Wikitext2 | FP | - | 14.62 | 12.47 | 10.86 | 10.12 |
|  | W4A16 | GPTQ | 15.56 | 12.82 | 11.41 | 10.31 |
|  |  | CBQ | **15.10** | **13.58** | **11.10** | **10.24** |
|  | W2A16 | OmniQuant | 23.95 | 18.13 | 14.43 | 12.94 |
|  |  | CBQ | **15.40** | **17.92** | **11.19** | **10.43** |

## J EVALUATION QUANTIZATION FOR LLAMA2 AND OPT ON W6A6

To further demonstrate the performance of the proposed CBQ, we conducted additional evaluations under the W6A6 setting on Llama2 and OPT models,as shown in Table 14.

Table 14: Evaluation quantization for LLAMA2 and OPT on W6A6

|  | #Bits | Methods | PIQA | HellaSwag | ARC-C | ARC-E | Mutual | Ethics | C4↓ | Wiki↓ |
|---|---|---|---|---|---|---|---|---|---|---|
| LLAMA2-7B | FP | - | 76.93 | 72.95 | 40.69 | 53.21 | 70.92/51.12/75.84 | 52.63 | 6.97 | 5.47 |
|  | W6A6 | Omniquant | 76.82 | 72.13 | 39.33 | 53.36 | 69.57/49.67/73.62 | 52.62 | 7.48 | 5.87 |
|  |  | CBQ | **77.58** | **72.14** | **40.27** | **53.87** | **70.16/50.22/74.83** | **53.02** | **7.24** | **5.67** |
| OPT-6.7B | FP | - | 76.49 | 67.18 | 34.64 | 60.14 | 69.02/47.85/74.71 | 57.65 | 11.74 | 10.86 |
|  | W6A6 | Omnquant | 75.89 | 66.73 | 33.61 | 60.05 | 67.95/46.16/73.70 | 55.95 | 11.81 | 10.96 |
|  |  | CBQ | **76.60** | **66.84** | **33.98** | **60.90** | **69.48/48.97/74.72** | **57.42** | **11.79** | **10.95** |

## K COARSE-TO-FINE PREPROCESSING ALGORITHM

Table 15: Ablation of the CFP for LLAMA-7B on W4A16.

| Bits | Method | C4↓ | Wiki↓ | PIQA | HellaSwag | ARC-C | ARC-E |
|---|---|---|---|---|---|---|---|
| W4A16 | CFP | 7.22 | 5.78 | 77.62 / 76.69 | 55.61 / 71.94 | 37.22 / 39.69 | 66.79 / 52.98 |
|  | CBD | 7.2 | 5.75 | 78.12 / 77.69 | 55.56 / 72.16 | 38.39 / 40.18 | 67.21 / 53.03 |

---

**Algorithm 1:** Coarse-to-Fine Preprocessing

---

**Input:** The input tensor $X$,
       The balancing coefficient $\lambda_1$, $\lambda_2$
**Output:** Outlier $O$

**1** **Coarse-grained Detection**;
**2** $X_{\text{sorted}} = \text{Sort}(X)$;
**3** $Q_1 = X[n/4], Q_3 = X[3n/4]$;
**4** $IQR = Q_3 - Q_1$;
**5** $T = Q_3 + \lambda_1 IQR$;
**6** $O = \{x | x > T, x \in X_{\text{sorted}}\}$;
**7** { **Fine-grained Detection**.}
**8** $N = \text{Len}(O), M^* = \textbf{INF}$;
**9** **foreach** $i = 0$ **to** $N$ **do**
**10**     $O_{outlier} = O_{i:N}$;
**11**     $O_{reserved} = O_{0:i}$;
**12**     $M_{intra} = \text{Var}(O_{reserved})$;
**13**     $M_{inter} = (\text{Min}(O_{outlier}) - \text{Max}(O_{reserved}))^2$;
**14**     $M = M_{inter} - \lambda_2 M_{intra}$;
**15**     **if** $M > M^*$ **then**
**16**         $O^* = O_{outlier}$;
**17**         $M^* = M$.
**18**     **end**
**19** **end**

---

