# OpenReview forum: "CBQ: Cross-Block Quantization for Large Language Models"
_ICLR.cc/2025/Conference — ICLR 2025 Spotlight_

### Official Review · Reviewer_mZis · 2024-10-29

**Soundness:** 3
**Presentation:** 3
**Contribution:** 2
**Rating:** 6
**Confidence:** 5

**Summary:**

This work introduces Cross-Block Quantization (CBQ), a novel post-training quantization (PTQ) framework targeting large language models (LLMs). CBQ's core innovation is its cross-block dependency mechanism, which introduces dependencies across multiple blocks to mitigate error accumulation. Additionally, a coarse-to-fine preprocessing strategy and LoRA-Rounding are proposed to handle weight and activation outliers. The authors validate CBQ on various LLMs and demonstrate that it achieves state-of-the-art performance in low-bit settings, like W4A4, across a range of datasets.

**Strengths:**

Innovation: The idea of using cross-block dependencies to reduce quantization errors is both intuitive and impactful, as it tackles a recognized issue in PTQ for LLMs.

Experimental Scope: CBQ is tested extensively across multiple LLM architectures and low-bit settings, showing clear performance gains over prior methods.

Significance: Reducing computational overhead for LLMs without retraining is valuable for real-world applications, and CBQ provides a viable solution for efficient model deployment.

**Weaknesses:**

Baseline Comparisons: Although the results on LLAMA models are promising, the exclusion of more recent PTQ methods like BiE[1] limits the impact of the results. Including such baselines could provide a more complete picture of CBQ’s effectiveness.

Broader Applicability: The focus on language models limits the generalizability of CBQ. Expanding evaluations to other architectures, such as vision transformers and VLMs (vision language models), could strengthen the paper’s applicability.

[1] BiE: Bi-Exponent Block Floating-Point for Large Language Models Quantization. ICML 2024

**Questions:**

The paper introduces a coarse-to-fine preprocessing (CFP) strategy for handling weight and activation outliers, yet it is unclear how this approach compares in complexity and execution time against other established methods like OMSE or SmoothQuant under similar quantization settings. Could the authors clarify the computational trade-offs of CFP compared to these methods? Additionally, a broader comparison in terms of both quantization accuracy and processing efficiency would strengthen the evaluation of CFP's effectiveness.

Can the authors elaborate on the expected quantization efficiency for models larger than 100 billion parameters?

---

> ### Author Response · Authors · 2024-11-16
>
> # **Rebuttal Revision Paper Modifications**
> We greatly appreciate your valuable review comments. We have revised the paper according to your suggestions and submitted the rebuttal version. For detailed modifications, please refer to the rebuttal version PDF and appendix L: Supplementary materials for rebuttal. Below, we address your identified weaknesses and questions, hoping to resolve your concerns.
>
> # **Weakness 1**
> We greatly appreciate your valuable review comments. We will include BIE in our baseline as part of the final version to provide a more complete picture of CBQ's effectiveness. Below is a comparison of our results with BIE, and we hope this helps clarify any confusion.
> |  | Method | #bits | piqa | arc_easy | Wiki |
> |---|---|---|---|---|---|
> | OPT-30B | FP | - | 78.18 | 65.40 | 9.56 |
> | OPT-30B | OmniQ | W4A4 | 75.38 | 61.25 | 10.60 |
> | OPT-30B | BiE | W4A4 | 75.45 | 61.34 | 10.52 |
> | OPT-30B | CBQ | W4A4 | 75.89 | 61.58 | 10.34 |
> | OPT-60B | FP | W4A4 | 79.81 | 67.26 | 9.34 |
> | OPT-60B | OmniQ | W4A4 | 77.85 | 63.29 | 10.29 |
> | OPT-60B | BIE | W4A4 | 77.96 | 63.45 | 9.96 |
> | OPT-60B | CBQ | W4A4 | 78.01 | 63.78 | 9.45 |
>
> # **Weakness 2**
> This is an interesting open-ended question, and in fact, we are also exploring this area. When moving beyond text-based LLMs and incorporating multimodal inputs, existing works like LLAVE often fine-tune the multimodal model (e.g., through methods like SFT) to help the LLM align with the multimodal inputs. We have found that further fine-tuning strengthens the inter-layer and intra-layer dependencies discussed in this paper, which provides us with additional research opportunities.

---

> ### Author Response · Authors · 2024-11-16
>
> # **Question 1**
> That's a great suggestion. Based on your feedback, we will expand our ablation experiments. Below are the final results, and we hope they help address your concerns.
> | Method | Time(min) | C4 | Wiki |
> |:---:|:---:|:---:|:---:|
> | W/o outlier pre-processing | 0 | 1082.68 | 1128.33 |
> | w/OMSE | 2 | 76.43 | 47.81 |
> | W/Percentile | 2 | 71.62 | 45.86 |
> | W/OS | 5 | 41.57 | 26.36 |
> | W/Smoothquant | 5 | 33.21 | 25.26 |
> | w/CFP-Act | 11 | 23.48 | 19.75 |
> | W/CFP-weight+CFP-Act | 15 | 21.98 | 17.95 |
>
> As the above table, the CFP coarse-to-fine preprocessing strategy introduces some additional computation time. However, this extra time is well worth it given the significant improvements in the final model quantization accuracy.
>
> # **Question 2**
>
> We evaluate the quantization efficiency of our weight-only CBQ quantization method and compare it to OmniQuant which is the representative reconstruction-based PTQ methods. The GPU training hours for both methods are shown in Table11 of appendix G. The results demonstrate that the training cost of CBQ can be faster than OmniQuant, particularly for larger models. For models with over 100 billion parameters, CBQ requires around 10 hours of training. However, for models of this massive scale, CBQ’s training efficiency is already very high compared to mainstream quantization methods. This indicates that our CBQ method offers an advantage in terms of training efficiency.
> | LLAMA | 7B | 13B | 30B | 65B |
> |:---:|:---:|:---:|:---:|---|
> | Omniquant | 1.1h | 2.2h | 4.5h | 8.9h |
> | CBQ | 0.9h | 1.45h | 2.1h | 4.3h |

---

### Official Review · Reviewer_FQxU · 2024-10-31

**Soundness:** 3
**Presentation:** 3
**Contribution:** 3
**Rating:** 6
**Confidence:** 4

**Summary:**

The authors find that optimizing for inter-layer dependencies is crucial, especially for lower-bit quantization. To address this challenge, they propose the use of cross-block loss where the loss is taken over a window of blocks and not just a single block. For the cross-block loss, in addition to using the L2-distance, the authors use KL-divergence between the activations. Inspired by AdaRound, the authors use LoRA parameters for accurate rounding. For outlier detection, the authors present a two step process. They use a quantile-based thresholding to identify an initial set of outliers following which they divide these coarse outliers into two sets by maximizing the inter-set variance and minimizing the intra-set variance.

**Strengths:**

- The paper clearly motivates the proposed cross-block quantization.
- CBQ shows substantial performance improvements over OmniQuant, especially in the 2-bit setting.
- The authors provide a thorough ablation study on several aspects of the proposed objective.

**Weaknesses:**

- The addition of LoRA parameters should take some hit in latency. The authors should present the latency numbers, especially for W2A16 and W4A4.
- Given that the paper focuses on two-bit quantization, comparison to several recent rotation-based methods such as FrameQuant [1] and QuIP [2] are omitted from the related works section and the results. The authors should compare against these methods for 2-bit quantization.
- In practice, most quantization frameworks use sub channel quantization for weights, the authors should report weight-only quantization numbers with sub-channel quantization for all bit-widths.

**Questions:**

- Majority of the experiments conducted in the paper are on the OPT and Llama-1 family of LLMs. It would be interesting to see some of the results presented in the paper on SOTA open-source LLMs like Llama-3 [3] and Gemma-2 [4].
- How exactly is W6A6 implemented, could the authors briefly discuss about its latency benefits by providing some numbers?

I am open to discussions and willing to reconsider my score.

[1] Adepu, Harshavardhan et al. “FrameQuant: Flexible Low-Bit Quantization for Transformers.” ICML 2024.

[2] Chee, Jerry et al. “QuIP: 2-Bit Quantization of Large Language Models With Guarantees.” NeurIPS 2023.

[3] Dubey, Abhimanyu et al. “The Llama 3 Herd of Models.” ArXiv abs/2407.21783 (2024): n. pag.

[4] Riviere, Gemma Team Morgane et al. “Gemma 2: Improving Open Language Models at a Practical Size.” ArXiv abs/2408.00118 (2024): n. pag.

---

> ### Author Response · Authors · 2024-11-16
>
> # **Rebuttal Revision Paper Modifications**
> We greatly appreciate your valuable review comments. We have revised the paper according to your suggestions and submitted the rebuttal version. For detailed modifications, please refer to the rebuttal version PDF and appendix L: Supplementary materials for rebuttal. Below, we address your identified weaknesses and questions, hoping to resolve your concerns and improve our score.
> # **Weakness 1**
> We apologize for any confusion caused by the phrasing in the paper. We will revise the wording accordingly. In contrast to LoRA-Adapters [1], LoRA-Rounding is not an independent parameter. During inference, it does not require separate computation like an adapter but is integrated directly into the model weights beforehand. As a result, it does not introduce any additional latency and GPU memory usage during inference. However, during training, it does result in increased GPU memory usage. We have shown the impact of LoRA-Rounding on GPU memory usage during the training phase in the ablation study. For your convenience, we have extracted this data into the table below:
>
> | Method | C4 | Wiki | #Epochs | GPU(GB) |
> |:---:|:---:|:---:|:---:|:---:|
> | w/o Rounding | 14.32 | 11.46 | 3 | 18.83 |
> | w/ Adarounding | 14.56 | 11.64 | 3 | 27.73 |
> | w/ Rounding | 13.86 | 10.98 | 3 | 27.73 |
> | w/ Rounding | 13.58 | 10.72 | 6 | 27.73 |
> | w/ LoRA-Rounding | 13.29 | 10.63 | 3 | 21.01 |
>
>  As observed，LoRA-Rounding leverages low-rank decomposition to reduce the number of learnable parameters and adjusts the training strategy, which not only decreases GPU memory consumption but also enhances training speed.
>
> # **Weakness 2**
> We greatly appreciate your valuable review comments. In the new version, we will include them as new baselines to demonstrate our performance on 2-bit quantization. Below are the comparison results between our method and theirs.
> | DataSet | Method | OPT-1.3B | OPT-2.7B | OPT-6.7B | OPT-13B |
> |---|---|---|---|---|---|
> | C4 | FP16 | 14.72 | 13.16 | 11.74 | 11.20 |
> | C4 | QUIP | 29,78 | 27.34 | 19.15 | 17.37 |
> | C4 | CBQ | 15.99 | 13.83 | 12.19 | 11.52 |
> | Wiki | FP16 | 14.62 | 12.47 | 10.86 | 10.12 |
> | Wiki | QUIP | 41.64 | 28.98 | 18.57 | 16.12 |
> | Wiki | FrameQuant | 30.54 | 20.67 | 15.72 | 13.45 |
> | Wiki | CBQ | 15.40 | 17.92 | 11.19 | 10.43 |
>
> # **Weakness 3**
> Thank you for your reminder. We will include a description of the weight-only quantization numbers with sub-channel quantization settings in the subsequent version of the paper. In this work, we have used 128 as the group size.

---

> ### Author Response · Authors · 2024-11-16
>
> # **Question 1**
> we are providing the experiments conducted on LLAMA3-7B below, hoping to clarify any confusion:
> | Method | Bits | PIQA | ARC-e | ARC-c | HellaSwag | C4 | WikiText2 |
> |:---:|:---:|:---:|:---:|:---:|:---:|:---:|:---:|
> |  | W16A16 | 79.9 | 80.1 | 50.4 | 60.2 | 9.2 | 6.1 |
> | RTN | W4A16 | 76.6 | 70.1 | 45.0 | 56.8 | 13.4 | 8.5 |
> | GPTQ | W4A16 | 78.4 | 78.8 | 47.7 | 59.0 | 10.4 | 6.5 |
> | AWQ | W4A16 | 79.1 | 79.7 | 49.3 | 59.1 | 9.4 | 6.6 |
> | QuIP | W4A16 | 78.8 | 75.8 | 46.9 | 57.4 | 9.6 | 7.1 |
> | Omniq | W4A16 | 79.2 | 79.6 | 49.5 | 59.2 | 9.6 | 6.8 |
> | CBQ | W4A16 | 79.5 | 80.6 | 50.1 | 59.7 | 9.2 | 6.3 |
> |Smoothquant | W8A8 | 79.1 | 79.5 |48.6  | 59.5 | 9.4 | 6.4|
> | Omniq | W8A8 |  79.5| 79.61 | 49.67 | 59.64 | 9.33 | 6.37 |
> | CBQ | W8A8 | 81.39| 80.97 | 50.65 | 60.03 | 9.3 |  6.28|
> | Omniquant | W4A8 | 78.62 |78.34  | 47.95 | 56.88 | 10.45 | 7.75 |
> | CBQ | W4A8 | 79.10 |79.12  | 48.65 | 57.98 | 10.12 |7.21  |
>
> # **Question 2**
> In LLM quantization, previous work [3-5] has predominantly used W6A6. This is primarily to represent the minimum bit-level quantization that can be achieved without significant loss in model performance. However, to achieve real acceleration, specific computation kernels need to support this type of quantization, which can bring about an acceleration factor of around 3x compared to W16A16.
>
> However, due to the unique bit representation of W6A6, there is significant memory fragmentation in practical kernels, requiring specialized engineering optimizations. As of now, there has been no practical implementation of a W6A6 computation kernel. This is why we have included this part of the experiment only in the appendix, as it serves mainly for a fair algorithmic comparison with prior work.
>
> For your concerns, I recommend looking into recent work such as T-MAC[2], which uses serialized bit representations. This approach can linearly enhance the acceleration effect across various bit-widths, achieving around 2.6x acceleration compared to W16A16. I hope this clarifies your confusion.
>
>
> [1]Hu E J, Shen Y, Wallis P, et al. Lora: Low-rank adaptation of large language models[J]. arXiv preprint arXiv:2106.09685, 2021.
>
> [2]Wei J, Cao S, Cao T, et al. T-mac: Cpu renaissance via table lookup for low-bit llm deployment on edge[J]. arXiv preprint arXiv:2407.00088, 2024.
>
> [3]Wei X, Zhang Y, Li Y, et al. Outlier suppression+: Accurate quantization of large language models by equivalent and optimal shifting and scaling[J]. arXiv preprint arXiv:2304.09145, 2023.
>
> [4]Shao W, Chen M, Zhang Z, et al. Omniquant: Omnidirectionally calibrated quantization for large language models[J]. arXiv preprint arXiv:2308.13137, 2023.
>
> [5]Liu J, Gong R, Wei X, et al. Qllm: Accurate and efficient low-bitwidth quantization for large language models[J]. arXiv preprint arXiv:2310.08041, 2023.

---

> > ### Comment · Reviewer_FQxU · 2024-11-17
> > **Official Comment**
> >
> > I thank the authors for the detailed response. I really appreciate the efforts the authors have put into addressing my concerns. I have increased my score to 6.

---

> > > ### Author Response · Authors · 2024-11-18
> > >
> > > We sincerely thank you for your support!

---

### Official Review · Reviewer_6n4w · 2024-10-31

**Soundness:** 3
**Presentation:** 3
**Contribution:** 3
**Rating:** 8
**Confidence:** 4

**Summary:**

This paper proposes Cross-Block Quantization (CBQ), a novel method designed to optimize post-training quantization (PTQ) for large language models (LLMs) by addressing inter-block dependencies and introducing a coarse-to-fine outlier handling strategy. CBQ's approach aims to preserve model accuracy at low-bit configurations by leveraging cross-block reconstructions and refining outlier quantization through LoRA-Rounding. The authors present experimental results demonstrating CBQ’s strong performance across different quantization bit widths and various LLM architectures, with promising efficiency for real-world deployment.

**Strengths:**

1. By focusing on inter-block dependencies, CBQ takes a proactive approach to minimize error accumulation, which is often a critical challenge in low-bit quantization. This dependency handling shows clear improvements in model accuracy.
2. The design of CBQ, particularly the coarse-to-fine outlier preprocessing and adaptive rounding, ensures flexibility across different LLM sizes, making it a practical choice for varied deployment needs.
3. The authors have included a wide range of experiments showing CBQ’s effectiveness in diverse quantization configurations, including settings that emphasize computational efficiency, which makes it relevant for both research and applied contexts.

**Weaknesses:**

1. While the paper demonstrates that overlapping blocks in CBQ contributes to performance, it lacks an in-depth analysis of how varying the overlap size impacts memory efficiency, latency, and overall quantization stability. Providing such details would clarify practical deployment considerations.
2. The coarse-to-fine preprocessing for outliers appears effective; however, an assessment of its necessity relative to simpler methods could be useful. It is unclear whether this specific strategy is essential or if comparable results could be achieved with simpler preprocessing.

**Questions:**

1. What would be the impact of reducing the overlap size in cross-block dependencies? Could reducing overlap compromise quantization efficiency, or are there scenarios where this would be advisable?
2. Could the authors clarify if the coarse-to-fine preprocessing provides unique benefits over standard outlier suppression methods? A comparative analysis would help isolate its specific contribution.
3. How do the authors envision CBQ scaling with models that incorporate multimodal inputs or models beyond text-based LLMs? Would additional adaptations be required?

---

> ### Author Response · Authors · 2024-11-16
>
> # **Rebuttal Revision Paper Modifications**
> We greatly appreciate your valuable review comments. We have revised the paper according to your suggestions and submitted the rebuttal version. For detailed modifications, please refer to the rebuttal version PDF and appendix L: Supplementary materials for rebuttal. Below, we address your identified weaknesses and questions, hoping to resolve your concerns.
> # **Weakness 1**
> We greatly appreciate your valuable review comments. Your feedback is highly important, We have provided the performance of CBD under different overlap and number of blocks, along with GPU memory usage and training time, in Appendix E to address your concerns. To make it easier for you to review, we have listed the results below. In future versions, we will highlight this experiment to draw the readers' attention more effectively:
> | #num of blocks | Overlap | C4 | Wiki | time(h) | GPU memory(GB) |
> |:---:|:---:|:---:|:---:|:---:|:---:|
> | 1 | 0 | 12.72 | 9.62 | 1.09 | 17.2 |
> | 2 | 0 | 12.56 | 9.34 | 1.50 | 21 |
> | 2 | 1 | 12.3 | 8.87 | 3.02 | 21 |
> | 4 | 0 | 12.34 | 8.89 | 1.1 | 39 |
> | 4 | 1 | 11.63 | 8.59 | 1.4 | 39 |
> | 4 | 2 | 11.42 | 8.28 | 1.96 | 39 |
> | 4 | 3 | 11.21 | 8.08 | 2.60 | 39 |
>
> Through this in-depth analysis, you will be able to make practical deployment considerations according to your specific needs.
>
> # **Weawkness 2**
> You are absolutely right. The coarse-to-fine preprocessing is just one solution strategy, and we hope to explore more straightforward methods in the future.
> In our ablation study, we provide comparison results both without using any strategy and when applying other existing strategies. The results are as follows:
> | Method | C4 | Wiki |
> |:---:|:---:|:---:|
> | W/o outlier pre-processing | 1082.68 | 1128.33 |
> | w/OMSE | 76.43 | 47.81 |
> | W/Percentile | 71.62 | 45.86 |
> | W/OS | 41.57 | 26.36 |
> | W/Smoothquant | 33.21 | 25.26 |
> | w/CFP-Act | 23.48 | 19.75 |
> | W/CFP-weight+CFP-Act | 21.98 | 17.95 |
> | w/OMSE + CBD | 25.34 | 19.53 |
> | W/Percentile + CBD | 25.62 | 19.45 |
> | W/OS + CBD | 17.83 | 13.89 |
> | W/smoothquant + CBD | 15.69 | 12.24 |
> | w/CFP-weight + Act + CBD | 13.29 | 10.63 |
>
> It can be observed that not applying any specific outlier strategy results in a significant degradation in quantization performance, with C4 PPL at 1082 and Wiki PPL at 1128. However, after applying outlier handling, CFP achieves the best results. We have also provided a visual comparison of the results before and after using CFP in Appendix F, where it is clear that CFP significantly improves the handling of outliers.
>
> So far, CFP has proven to be an effective approach. While we believe that simpler and more efficient methods will emerge in the future, the design principles behind CFP can still inspire future researchers (please refer to Q2 for more details).

---

> ### Author Response · Authors · 2024-11-16
>
> # **Question1**
> You are correct. To improve quantization performance, it is necessary to increase the overlap. However, this increase also results in longer training times. Increasing the number of blocks can reduce training time to some extent, but it also increases GPU memory usage. Therefore, in this paper, to achieve a balance, we use O3N4 for extreme cases:W2A16, while for all other cases, we use the O1N2 configuration. We will include this clarification in the updated version of the paper.
> | #num of blocks | Overlap | C4 | Wiki | time(h) | GPU memory(GB) |
> |:---:|:---:|:---:|:---:|:---:|:---:|
> | 1 | 0 | 12.72 | 9.62 | 1.09 | 17.2 |
> | 2 | 0 | 12.56 | 9.34 | 1.50 | 21 |
> | 2 | 1 | 12.3 | 8.87 | 3.02 | 21 |
> | 4 | 0 | 12.34 | 8.89 | 1.1 | 39 |
> | 4 | 1 | 11.63 | 8.59 | 1.4 | 39 |
> | 4 | 2 | 11.42 | 8.28 | 1.96 | 39 |
> | 4 | 3 | 11.21 | 8.08 | 2.60 | 39 |
>
> # **Question 2**
> Thank you very much for your suggestions. We will isolate the specific contribution in the upcoming revisions. Below are some of our analyses about existing outlier process methods, which we hope will address your concerns:
>
> 1.They only focus on handling activation, even transferring outliers from activations to weights, which makes it challenging to quantize the weights.
>
> 2.They often assume that data follow a normal distribution, which is not always strictly applicable to real-world datasets，The relevant theory can be found in Appendix F.
>
> 3.They lack a precise method for detecting outliers in activation channels in LLM quantization. This deficiency can result in damage to normal activation channels and weights from detection errors.
>
> Our method(CFP) differs from them: (1) We decoupled the outlier processing in activations and weights. (2) We avoid assuming specific data distributions: First, we use coarse-grained detection based on quartile statistics[1] to process per-channel statistics. (3)Then, we apply fine-grained detection using our proposed distance metric that effectively measures the intra-class and inter-class distances between outlier and normal values.
>
>
> # **Question 3**
> This is an interesting open-ended question, and in fact, we are also exploring this area. When moving beyond text-based LLMs and incorporating multimodal inputs, existing works like LLAVE often fine-tune the multimodal model (e.g., through methods like SFT) to help the LLM align with the multimodal inputs. We have found that further fine-tuning strengthens the inter-layer and intra-layer dependencies discussed in this paper, which provides us with additional research opportunities.
>
> [1]Bland M. Estimating mean and standard deviation from the sample size,three quartiles,minimum,and maximum[J].International Journal of Statistics in Medical Research, 2015, 4(1): 57-64.

---

> > ### Comment · Reviewer_6n4w · 2024-12-03
> >
> > Based on the author's thorough response and revisions, I raise the score to 8.

---

### Official Review · Reviewer_s3vr · 2024-11-03

**Soundness:** 4
**Presentation:** 4
**Contribution:** 4
**Rating:** 10
**Confidence:** 5

**Summary:**

This paper introduced CBQ  post-training quantization PTQ technique designed for compressing large language models under low-bit precision. The authors identified the primary challenge in ultra-low-bit quantization: the dependencies within and between layers that amplify quantization errors, especially as model size and parameter counts increase. To address this, CBQ incorporates a cross-block reconstruction strategy, which leverages both intra-layer and inter-layer dependencies by optimizing multiple transformer blocks within a sliding window approach. Additionally, the method employs a LoRA-Rounding technique to manage intra-layer dependencies and reduce computational costs, while an adaptive coarse-to-fine preprocessing strategy effectively handles outliers in weights and activations.

**Strengths:**

I have carefully read this work, I think the result of this work was convincing, and the approach was solid.

**Weaknesses:**

N/A

**Questions:**

I have read this paper multiple times before, and I fully agree with the authors' approach. I have no further questions.

---

> ### Author Response · Authors · 2024-11-16
>
> We sincerely thank you for your positive evaluation and the perfect score. Your comments of our work is deeply encouraging, and we truly appreciate your valuable insights. If you have any further suggestions or would like to discuss our work in more detail, we would be delighted to engage in a conversation. Your feedback is invaluable to us. Thank you again for your support!

---

### Official Review · Reviewer_AGza · 2024-11-04

**Soundness:** 3
**Presentation:** 3
**Contribution:** 3
**Rating:** 8
**Confidence:** 3

**Summary:**

The paper introduces Cross-Block Quantization (CBQ), a novel post-training quantization (PTQ) technique developed for large language models (LLMs). CBQ tackles critical inter- and intra-layer dependencies that compromise quantization accuracy at ultra-low bit settings, establishing long-range dependencies through cross-block reconstruction and managing intra-layer dependencies via adaptive LoRA-Rounding. The approach also incorporates a coarse-to-fine preprocessing method to optimize handling of weights and activations.

**Strengths:**

1. The paper offers an interesting insight: full model quantization introduces inter-layer correlations.

2. Experimental results on LLAMA1, LLAMA2, and OPT demonstrate substantial improvements in W2 and W4 settings, alongside a significant reduction in PTQ scenario.

**Weaknesses:**

1. The writing could be improved, especially regarding clarity in notation. While the concepts are interesting, it is difficult to follow due to unclear notation. For example, symbols like \(i\), \(j\), and \(K\) in Equation 3, as well as \(V\) in Equation 11, are not clearly defined when first introduced. It would help if each symbol had an explicit definition upon first use. Additionally, the term "scales" in the phrase "comparisons of the scales between adjacent layers..." lacks clarity. Specifying what "scales" refers to in the context of adjacent layers would enhance readability.

2. The range of models tested is limited. While the authors include results on LLAMA1, LLAMA2, and OPT, these are relatively dated models, and results on newer models like LLAMA3, Mistral, and Falcon should be included.

3. While the authors suggest that LoRA-Rounding was introduced to reduce computation, they do not evaluate this aspect in their experiments. To make this claim more compelling, the authors could include specific metrics or experiments comparing the computational efficiency of LoRA-Rounding to alternative approaches.

4. The authors should compare their method with other W4 approaches, such as AWQ. Comparing with AWQ would be valuable because of its relevant strengths or similarities, which could provide a more comprehensive evaluation of the method’s performance in a competitive context.

**Questions:**

1. Will the code be released?

2. LoRA-Rounding needs clearer explanation. Is *V* the rounding mask? On what type of matrix is it applied, and does each matrix in the model have its own *V*?

3. Has CBQ been tested for improvements in few-shot scenarios?

4. Figure 1 is intriguing; does this phenomenon appear in other models, like LLAMA3, Mistral, or Falcon?

---

> ### Author Response · Authors · 2024-11-16
>
> # **Rebuttal Revision Paper Modifications**
> We greatly appreciate your valuable review comments. We have revised the paper according to your suggestions and submitted the rebuttal version. For detailed modifications, please refer to the rebuttal version PDF and appendix L: Supplementary materials for rebuttal. Below, we address your identified weaknesses and questions, hoping to resolve your concerns.
> # **Weakness 1**
> We apologize for any confusion caused by the phrasing in the paper. Your feedback is highly important, and we will revise our statements as follows:
>
> 1.	We can clarify the symbol definitions with a more intuitive explanation. Here, we are interested in exploring the impact of $K$ elements on the final quantization error. Using the Hessian matrix, we store the relationship between any two elements $i$ and $j$, leading to the expression in Equation (3). We have visualized the results in Figure 1, where we observe a notable increase in the values of off-diagonal elements during lower-bit quantization.
>
> 2.	Here, $V$ represents the weight-compensation matrices, with each matrix in the model having its own. We will revise the expression to make this clearer.
>
> 3.	The term "scale" here refers to the quantization step size. As shown in Figure 1, experiments indicate that for low-bit quantization, the optimal quantization step size of a given layer should be determined with consideration of the step sizes of its neighboring layers.
>
> # **Weakness 2**
> Thank you very much for your suggestion.  We conducted experiments on the latest LLAMA3 model to address and clarify your concerns.
> | Method | Bits | PIQA | ARC-e | ARC-c | HellaSwag | C4 | WikiText2 |
> |:---:|:---:|:---:|:---:|:---:|:---:|:---:|:---:|
> |  | W16A16 | 79.9 | 80.1 | 50.4 | 60.2 | 9.2 | 6.1 |
> | RTN | W4A16 | 76.6 | 70.1 | 45.0 | 56.8 | 13.4 | 8.5 |
> | GPTQ | W4A16 | 78.4 | 78.8 | 47.7 | 59.0 | 10.4 | 6.5 |
> | AWQ | W4A16 | 79.1 | 79.7 | 49.3 | 59.1 | 9.4 | 6.6 |
> | QuIP | W4A16 | 78.8 | 75.8 | 46.9 | 57.4 | 9.6 | 7.1 |
> | Omniq | W4A16 | 79.2 | 79.6 | 49.5 | 59.2 | 9.6 | 6.8 |
> | CBQ | W4A16 | 79.5 | 80.6 | 50.1 | 59.7 | 9.2 | 6.3 |
> |Smoothquant | W8A8 | 79.1 | 79.5 |48.6  | 59.5 | 9.4 | 6.4|
> | Omniq | W8A8 |  79.5| 79.61 | 49.67 | 59.64 | 9.33 | 6.37 |
> | CBQ | W8A8 | 81.39| 80.97 | 50.65 | 60.03 | 9.3 |  6.28|
> | Omniquant | W4A8 | 78.62 |78.34  | 47.95 | 56.88 | 10.45 | 7.75 |
> | CBQ | W4A8 | 79.10 |79.12  | 48.65 | 57.98 | 10.12 |7.21  |
>
> # **Weakness 3**
> Thank you very much for your suggestion.  In our ablation study, we compared LoRA-Rounding with AdaRound and Rounding in terms of training epochs, GPU memory consumption, and final performance. The results show that LoRA-Rounding achieves superior performance in just 3 epochs, outperforming Rounding at 6 epochs, while also requiring less GPU memory. This demonstrates that LoRA-Rounding leverages low-rank decomposition to reduce the number of learnable parameters and adjusts the training strategy, which not only decreases GPU memory consumption but also enhances training speed.
> | Method | C4 | Wiki | #Epochs | GPU(GB) |
> |:---:|:---:|:---:|:---:|:---:|
> | w/o Rounding | 14.32 | 11.46 | 3 | 18.83 |
> | w/ Adarounding | 14.56 | 11.64 | 3 | 27.73 |
> | w/ Rounding | 13.86 | 10.98 | 3 | 27.73 |
> | w/ Rounding | 13.58 | 10.72 | 6 | 27.73 |
> | w/ LoRA-Rounding | 13.29 | 10.63 | 3 | 21.01 |
>
> # **Weakness 4**
> Thank you very much for your suggestion. In future versions, we will include AWQ as a baseline for W4. Below is a comparison across various model sizes for LLAMA and LLAMA2, with the results shown as follows:
>
> | WikiText2 | L2-7B | L2-13B | L2-70B | L-7B | L-13B | L-30B | L-65B |
> |:---:|:---:|:---:|:---:|:---:|:---:|:---:|:---:|
> | FP16 | 5.47 | 4.88 | 3.32 | 5.68 | 5.09 | 4.10 | 3.53 |
> | GPTQ | 5.69 | 4.98 | 3.42 | 6.22 | 5.23 | 4.24 | 3.66 |
> | AWQ | 5.60 | 4.97 | 3.41 | 5.78 | 5.19 | 4.21 | 3.62 |
> | CBQ | 5.57 | 4.95 | 3.38 | 5.73 | 5.15 | 4.18 | 3.59 |

---

> ### Author Response · Authors · 2024-11-16
>
> # **Question 1**
> The code will be released after the paper submission, and we look forward to further discussions with you.
> # **Question 2**
> Please refer to the response to Weakness1.
> # **Question 3**
> We conducted a completely fair comparison of zero-shot and generation tasks, as shown in Tables 1 and 2. Based on your request, we provide few-shot results on the MMLU-pro dataset using LLaMA3-8B under 5-shot settings with W8A8 and W4A8 precision.
> |  | #bit | Avg | STEM | other | Social Science | Humanities |
> |---|:---:|:---:|:---:|:---:|:---:|---|
> | FP16 | FP16 | 61.26 | 50.15 | 69.01 | 73.88 | 54.63 |
> | W8A8 | omniq | 60.6 | 48.12 | 68.45 | 73.29 | 54.45 |
> | W8A8 | CBQ | 60.87 | 49.59 | 68.85 | 73.59 | 55.21 |
> | W4A8 | omniq | 56.10 | 45.17 | 64.50 | 67.65 | 48.85 |
> | W4A8 | CBQ | 57.63 | 45.57 | 64.81 | 68.54 | 49.32 |
> # **Question 4**
> In fact, prior to submitting the paper, we conducted the same experiments on LLAMA, LLAMA2, OPT, and BLOOM, and observed a significant increase in both inter-layer and intra-layer dependencies. Following your suggestion, we applied the same analysis to LLAMA3, and the results were consistent with the phenomena described in the paper. We have included the LLAMA3 and OPT visualizations in the appendix, which can be found in Appendix L of rebuttal version.

---

> > ### Comment · Reviewer_AGza · 2024-11-28
> >
> > Thanks for your answer.

---

> ### Comment · Reviewer_AGza · 2024-11-28
>
> Thank you for your detailed explanation and experiments, which have addressed most of my concerns. I raised my score accordingly.

---

### Meta-Review · Area_Chair_UN4W · 2024-12-21

**Metareview:**

The reviewers listed some weaknesses: The evaluation is limited to older models like LLAMA1, LLAMA2, and OPT, with no comparison to newer models like LLAMA3, Mistral, or Falcon. Despite claims that LoRA-Rounding reduces computation, no experiments are provided to support this. The comparison with other methods, such as AWQ, is missing, and the impact of varying overlap size in CBQ is not analyzed in depth, especially regarding memory efficiency and latency. Additionally, the necessity of the coarse-to-fine preprocessing strategy is unclear, and latency measurements for certain bit-widths are absent. Comparisons to recent rotation-based methods like FrameQuant and QuIP, as well as weight-only quantization with sub-channel quantization, are also omitted. Finally, the focus on LLAMA models limits the generalizability of the approach, and including evaluations on other architectures like vision transformers or VLMs would broaden its applicability.
And the authors addressed most of them in the rebuttal period and reviewers raised their rating. One of the reviews that was strong accept was very short, so it is not heavily considered in this decision. There is no negative review, so will be accepted.

**Additional Comments On Reviewer Discussion:**

The authors discussed with 3 reviewers and one of the reviewers did not respond. In the end 3 reviewers raised their rating after reading the author's response.

---

### Decision · Program_Chairs · 2025-01-22

Accept (Spotlight)